# Hybrid Nanoparticles of Proanthocyanidins from *Uncaria tomentosa* Leaves: QTOF-ESI MS Characterization, Antioxidant Activity and Immune Cellular Response

**DOI:** 10.3390/plants11131737

**Published:** 2022-06-30

**Authors:** Andrea Mariela Araya-Sibaja, Krissia Wilhelm-Romero, Felipe Vargas-Huertas, María Isabel Quirós-Fallas, Diego Alvarado-Corella, Juan José Mora-Román, José Roberto Vega-Baudrit, Andrés Sánchez-Kopper, Mirtha Navarro-Hoyos

**Affiliations:** 1Laboratorio Nacional de Nanotecnología LANOTEC-CeNAT-CONARE, Pavas 1174-1200, Costa Rica; krissia.wilhelm@ucr.ac.cr (K.W.-R.); jvegab@gmail.com (J.R.V.-B.); 2Sede Central, Universidad Técnica Nacional, Alajuela 159-7050, Costa Rica; 3Laboratorio BIODESS, Escuela de Química, Universidad de Costa Rica, San José 2060, Costa Rica; luis.vargashuertas@ucr.ac.cr (F.V.-H.); maria.quirosfallas@ucr.ac.cr (M.I.Q.-F.); luis.alvaradocorella@ucr.ac.cr (D.A.-C.); mnavarro@codeti.org (M.N.-H.); 4Facultad de Farmacia, Universidad de Costa Rica, San José 2060, Costa Rica; juanjose.moraroman@ucr.ac.cr; 5Laboratorio de Investigación y Tecnología de Polímeros POLIUNA, Escuela de Química, Universidad Nacional de Costa Rica, Heredia 86-3000, Costa Rica; 6Centro de Investigación y de Servicios Químicos y Microbiológicos (CEQIATEC), Escuela de Química, Tecnológico de Costa Rica, Cartago 159-7050, Costa Rica; ansanchez@itcr.ac.cr

**Keywords:** *Uncaria tomentosa*, mass spectroscopy, hybrid nanoparticles, proanthocyanidins, antioxidant activity, delayed-type hypersensitive test

## Abstract

Previous studies in *Uncaria tomentosa* have shown promising results concerning the characterization of polyphenols with leaves yielding more diverse proanthocyanidins and higher bioactivities values. However, the polyphenols-microbiota interaction at the colonic level and their catabolites avoid the beneficial effects that can be exerted by this medicinal plant when consumed. In this regard, a new generation of hybrid nanoparticles has demonstrated improvements in natural compounds’ activity by increasing their bioavailability. In this line, we report a detailed study of the characterization of a proanthocyanidin-enriched extract (PA-E) from *U. tomentosa* leaves from Costa Rica using UPLC-QTOF-ESI MS. Moreover, two types of hybrid nanoparticles, a polymeric-lipid (F-1) and a protein-lipid (F-2) loaded with PA-E were synthesized and their characterization was conducted by dynamic light scattering (DLS), attenuated total reflectance Fourier transform infrared spectroscopy (ATR-FT-IR), high-resolution transmission electron microscopy (HR-TEM), and encapsulation efficiency (%EE). In addition, in vitro release, antioxidant activity through 2,2-diphenyl-1-picrylhidrazyl (DPPH) as well as in vivo delayed-type hypersensitivity (DTH) reaction was evaluated. Results allowed the identification of 50 different compounds. The PA-E loaded nanoparticles F-1 and F-2 achieved encapsulation efficiency of ≥92%. The formulations exhibited porosity and spherical shapes with a size average of 26.1 ± 0.8 and 11.8 ± 3.3 nm for F-1 and F-2, respectively. PA-E increased its release rate from the nanoparticles compared to the free extract in water and antioxidant activity in an aqueous solution. In vivo, the delayed-type hypersensitive test shows the higher immune stimulation of the flavan-3-ols with higher molecular weight from *U. tomentosa* when administered as a nanoformulation, resulting in augmented antigen-specific responses. The present work constitutes to our knowledge, the first report on these bioactivities for proanthocyanidins from *Uncaria tomentosa* leaves when administrated by nanosystems, hence, enhancing the cellular response in mice, confirming their role in immune modulation.

## 1. Introduction

The Uncaria genus has been instrumental in the discovery of medicinal natural products [1]. A wide range of recorded traditional medicine applications of *Uncaria tomentosa* (*U. tomentosa*) (Willd. ex Schult.) DC., commonly known as ‘cat’s claw’, includes blood purification, anticoagulant, haemorrhage therapy, anti-tumor, anti-inflammatory, to treat rheumatism, arthritis, gastrointestinal disorders, viral infections (including AIDS), skin impurities, contraceptive, anti-parasitic, and antioxidant activities [2,3]. The efficacy of cat’s claw was originally attributed to the presence of oxindole alkaloids; however, recent intensive research has driven attention to the medicinal properties of the polyphenols present in *U. tomentosa* [4,5].

Our previous studies in *U. tomentosa* have shown promising results regarding the characterization of polyphenols with leaves yielding more diverse proanthocyanidins and higher bioactivities values [6,7]. From the analysis of *U. tomentosa* leaves’ polyphenolic-enriched extracts and compounds identification [4,8], it becomes clear that besides alkaloids [1,9] flavan-3-ols are the main compounds, comprising procyanidins, propelargonidins and flavalignans.

Proanthocyanidins and their monomeric units, catechin and epicatechin are some of the most flavonoid types sought; for instance, they are regarded as functional ingredients in various beverages, whole and processed foods, herbal remedies and supplements [10]. However, from all the variety of polyphenols currently discovered, flavan-3-ols, often referred to as flavanols, are the most complex class of flavonoids because they range in size from simple monomers to oligomeric and polymeric proanthocyanidins [11], which are also known as condensed tannins.

Flavan-3-ols found in *U. tomentosa* have different degrees of hydroxylation on the B-ring, which is a key feature for their discrimination; while procyanidins have hydroxyl groups at the 3′ and 4′ positions, propelargonidins have a single -OH group at the 4′ position [12]. The flavan-3-ol structure determines the relative ease of oxidation and free radical-scavenging activity, which is primarily attributed to the high reactivity of hydroxyl substituents. Thus, an increasing degree of polymerization enhances the effectiveness of procyanidins due to the increment of the hydroxyls in addition to the extensive conjugation between 3-OH in the A-ring and the B-ring hydroxyl groups [13]. Unlike other flavonoids, flavan-3-ols are generally found in free rather than glycosylated or esterified form [14].

About 90% of the oligomeric proanthocyanidins (PA) intake reaches the colon under physiological circumstances, where they become available for colonic biotransformation by resident microflora, which can generate several phenolic acids and other smaller aromatic compounds more readily absorbed in situ. Therefore, the various health benefits ascribed to PA might be related not to the direct actions of oligomers themselves, but to the actions of their colon-derived metabolites [15]. In fact, colonic metabolites are highly bioavailable in the human organism [16] and since the catabolites generated by microbiota processes are linked to the structure of their monomer subunits, the presence of such diversity of flavan-3-ols in *U. tomentosa* makes it an outstanding candidate as a nutraceutical and therapeutical agent.

Besides procyanidins, which have been intensively studied, the literature regarding the bioactivity of other flavan-3-ols, such as propelargonidins and flavalignans, is more limited. In our previous studies, propelargonidin dimers were of particular interest, showing a significant correlation with antioxidant activities [8] and cytotoxic selectivity on gastric AGS and colon SW620 adenocarcinoma cell lines [6].

Comparing the anti-inflammatory effects of propelargonidin monomers, namely (epi)afzlechin, some authors have found that they are capable of inhibiting the accumulation of pro-inflammatory inducible nitric oxide synthase and cyclooxygenase-2 proteins in TNF-a-stimulated HepG2 cells. For instance, afzelechin is identified as a good inhibitor of the TNF-a-induced NF-kB activation in HepG2 cells, however, its glycosylated form and its cis stereoisomer epiafzelechin were less efficient [17], stretching the importance of different bioactivities depending on substituted aglycones. In silico studies showed that propelargonidin molecules are able to bind strongly to the active site of the macromolecule of the tyrosinase enzyme [18], which can in turn be a good lead molecule to target depigmentation disorders in humans [19].

Flavalignans, such as the ones reported in *U. tomentosa*, on the other hand, have shown antioxidant and antimicrobial activity [20,21]. Among flavalignans, cinchonain has also shown an insulinotropic effect in vitro and in vivo. In addition, cinchonain increased insulin levels in vitro to a similar degree that glibenclamide, a drug that stimulates insulin release and also increased plasma insulin levels in vivo in normal rats [22].

Since the food matrix modifies the polyphenols-microbiota interaction at the colonic level and therefore their catabolites [15], methodologies to increase in vivo biotransformation of such compounds are crucial to formulating products with the highest bioactivity. In light of this background, the loading of natural products into nanosystems has been widely studied since nanoparticles can optimize beneficial properties that improve the activity of natural compounds and extracts by increasing their bioavailability [23]. Within the recent nano-based strategies, there highlights a new generation of hybrid nanoparticles that have emerged as a combination of a polymeric or protein core and lipid shell, performing a controlled release to increase in vivo biotransformation of these bioactives [24,25].

In this contribution, we performed a detailed characterization of a proanthocyanidin-enriched extract (PA-E) from *U. tomentosa* leaves from Costa Rica using UPLC-QTOF-ESI MS to synthesize two different types of hybrid (i.e., one polymeric-lipid and one protein-lipid) PA-E loaded nanoparticles and characterize them through DLS, FT-IR, TEM, and encapsulation efficiency. In addition, in vitro release and antioxidant activity through 2,2-diphenyl-1-picrylhidrazyl (DPPH) as well as in vivo delayed-type hypersensitivity (DTH) reaction were evaluated on the enriched extract and both nanoformulations.

## 2. Results

### 2.1. Profile by UPLC-DAD-ESI-MS/MS Analysis

The analysis of UPLC-DAD-ESI-MS/MS described in the materials and methods section, allowed the identification of 50 different flavan-3-ols in the enriched proanthocyanidins extract PA-E from *U. tomentosa* leaves from Costa Rica, including the two monomers, epicatechin and catechin, four procyanidin dimers conformed by (epi)catechin units, four propelargonidin dimers with (epi)afzelechin units and four propelargonidin dimers formed by (epi)catechin and (epi)afzelechin monomers; as well as seven procyanidin trimers composed of (epi)catechin monomers, six properlagonidin trimers with (epi)afzelechin units, eight propelargonidin trimers formed by two units of (epi)afzelechin and one unit of (epi)catechin, eight properlagonidin trimers with one unit of (epi)afzelechin and two (epi)catechin monomers, and finally, seven flavalignans. Table 1 summarizes the analysis results for these compounds and Figure 1 and Figure 6 summarize the chromatograms for all 50 compounds.

The flavan-3-ols monomers catechin and epicatechin (Figure 2) were tentatively assigned to peaks 14 (Rt = 12.83 min) and 30 (Rt = 17.24 min). Both peaks show a [M-H]^−^ at *m*/*z* 289.0729 (C_15_H_13_O_6_) with main fragments *m*/*z* 245 [M-H-42]^-^ due to retro-Diels-Alder fission (RDA) of ring A, and *m*/*z* 179 by loss of ring B [26].

Four procyanidin B-type dimers are present in peaks 4 (Rt = 9.58 min), 7 (Rt = 10.38 min), 13 (Rt = 12.70 min), and 17 (Rt = 13.84 min), which show [M-H]^−^ at *m*/*z* 577.1382 (C_30_H_25_O_12_). As shown in Figure 3, the fragment at *m*/*z* 559 originates from water loss (18 Da) and the ion at *m*/*z* 451 results from the elimination of the phloroglucinal (126 Da) by heterocyclic ring fission (HRF). The fragment ion at *m*/*z* 425 [M-H-152]^−^ comes from retro-Diels−Alder (RDA) and the subsequent loss of water gives the fragment at *m*/*z* 407, while the ion at *m*/*z* 289 originates from quinone methide (QM) fission, resulting in the monomer ion [27,28].

Four propelargonidin dimers, formed by two (epi)-afzelechins subunits, are present in peaks 23 (Rt = 15.79 min), 25 (Rt = 16.18 min), 36 (Rt = 18.24 min) and 40 (Rt = 19.71 min), which show [M-H]^−^ at *m*/*z* 545.1473 (C_30_H_25_O_10_). The main fragments (Figure 4) at *m*/*z* 419 [M–H-126]^−^ are due to the loss of ring A during the HRF fragmentation, and ions at *m*/*z* 273 and 271 come from the QM cleavage of the interflavan bond. The fragment at *m*/*z* 273 can also be produced through the RDA fragmentation pathway in both subunits [26].

Four propelargonidin mixed dimers composed of (epi)afzelechin and (epi)catechin units are present in peaks 11 (Rt = 12.62 min), 16 (Rt = 13.68 min), 22 (Rt = 15.63 min) and 32 (Rt = 17.50 min), which show [M-H]^-^ at *m*/*z* 561.1416 (C_30_H_25_O_11_). As shown in Figure 5, the main fragments were at *m*/*z* 435 [M–H-126]^-^ due to the loss of ring A during the HRF pathway; the RDA fragmentation pathway generates product ions at *m*/*z* 425 and 407, and QM generates ions at *m*/*z* 273 and 289 [26,28].

Four types of B-type trimers were found in *U. tomentosa* leaves proanthocyanidin-enriched extract, whose chromatograms are summarized in Figure 6.

On one hand, seven procyanidin B-type trimers were found in peaks 1 (Rt = 5.09 min), 6 (Rt = 10.35 min), 8 (Rt = 10.77 min), 9 (Rt = 11.55 min), 12 (Rt = 12.67 min), 18 (Rt = 14.39 min), and 26 (Rt = 16.30 min), exhibiting [M-H]^−^ at *m*/*z* 865.2084 (C_45_H_37_O_18_). Their skeleton is composed of three (epi)catechin subunits linked through 4β-8C, and suffer QM cleavage of the upper interflavanoid bond producing ions of *m*/*z* 287 and 577 (Figure 7), whereas cleavage of the lower interflavanoid bond forms main fragment ions at *m*/*z* 289 and *m*/*z* 575 [29].

Six pure afzelechin B-trimers formed by three (epi)-afzelechins subunits correspond to peaks 5 (Rt = 10.01 min), 29 (Rt = 17.15 min), 37 (Rt = 18.57 min), 39 (Rt = 19.42 min), 43 (Rt = 21.84 min), and 44 (Rt = 22.79 min), which show [M-H]^−^ at *m*/*z* 817.2179 (C_45_H_37_O_15_). The main fragments (Figure 8) are present at *m*/*z* 799 [M–H-18]^−^ due to the loss of water, *m*/*z* 545 due to QM cleavage of the upper interflavanoid bond, indicating that the mid and base units have B-type linkage. Additional fragments are present at *m*/*z* 527 (loss of water) and *m*/*z* 273 (QM cleavage) [28]. Other previously reported fragments show at *m*/*z* 665 due to HRF of the lower unit, and *m*/*z* 637 that occurs by RDA and the additional loss of C_2_H_3_OH (44 Da) [26].

Eight propelargonidin B-type trimers with two (epi)afzelechin subunits and one (epi)catechin subunit were found in peaks 3 (Rt = 7.33 min), 19 (Rt = 14.56 min), 21 (Rt = 15.57 min), 27 (Rt = 16.36 min), 28 (Rt = 16.58 min), 33 (Rt = 17.56 min), 34 (Rt = 18.13 min), and 41 (Rt = 20.91 min), showing [M-H]^−^ at *m*/*z* 833.2141 (C_45_H_37_O_16_). The main fragments show at *m*/*z* 561 due the (epi)afzelechin moiety loss, which originates fragments at *m*/*z* 543 (loss of water), *m*/*z* 435 (HRF pathway) and *m*/*z* 289 due to the loss of the catechin moiety (Figure 9) [28].

Another eight propelargonidin B-type trimers composed of one (epi)afzelechin subunit and two (epi)catechin subunits were found in peaks 2 (Rt = 6.04 min), 10 (Rt = 12.29 min), 15 (Rt = 13.63 min), 20 (Rt = 15.02 min), 24 (Rt = 15.89 min), 31 (Rt = 17.45 min), 35 (Rt = 18.21 min), and 38 (Rt = 19.27 min), showing [M-H]^−^ at *m*/*z* 849.2089 (C_45_H_37_O_17_). The main fragment ions show at *m*/*z* 723 due to the loss of the A-ring by HRF, *m*/*z* 697 due to RDA on one (epi)catechin subunit (Figure 10). The ion peaks at *m*/*z* 561 and 559 are associated with an (epi)afzelech-(epi)catechin fragment due to the QM pathway, producing additional fragments at *m*/*z* 407 due to RDA on the (epi)catechin subunit, and *m*/*z* 289 and 273 as a result of QM, confirming the presence of (epi)catechin and (epi) afzelechin moieties [30].

Finally, seven flavalignans were found in peaks 42 (Rt = 21.61 min), 45 (Rt = 23.22 min), 46 (Rt = 26.06 min), 47 (Rt = 26.85 min), 48 (Rt = 27.94 min), 49 (Rt = 28.47 min), and 50 (Rt = 32.76 min), showing [M-H]^−^ at *m*/*z* 451.1039 (C_24_H_19_O_9_). These peaks were tentatively assigned to cinchonain derivates, with main fragments (Figure 11) at *m*/*z* 341 due to the loss of the catechol unit and *m*/*z* 289 due to the loss of [C_3_H_2_O]^−^ [31].

### 2.2. ^13^C-NMR Analysis of *U. tomentosa* Leaves Polyphenolic Extract

In agreement with the UPLC-QTOF-ESI MS results, the ^13^C-NMR analysis of the *U. tomentosa* leaves extract showed signals characteristic of procyanidins and propelargonidins as previously reported [4,8]. For instance, the ^13^C-NMR spectra (Figure 12) show signals between δ 160 and 155 ppm that can be attributed to carbon C4’ of the B-ring for propelargonidins (PP) and to carbons C5, C7 of the A ring, as illustrated in their structures (Figure 2) from both, procyanidins (PC) and propelargonidins (PP) [32,33].

The chemical shifts for characteristic B ring resonances appear in the range of δ 150–115 ppm, serving to differentiate a specific type of compound. For instance, signals at δ 145.6 ppm (C3′ and C4′ of the B ring), δ 132.1 ppm (C1′ of the B ring) and δ 119.4 ppm (C6′ of the B ring) are characteristic of procyanidins (PC). On the other hand, signals at δ 130.5 ppm (C1′ of the B ring) and at δ 129.2 ppm (C2′ and C6′ of the B ring) are distinctive for propelargonidins (PP). Meanwhile, the cluster of peaks between δ 117–115 ppm corresponds to C2′, C5′ (PC) and C3′, C5′ (PP).

The upfield shifts from B ring resonances corresponding to signals to the right of 115 ppm provided important structural information. On one hand, the stereochemistry of the C ring showing signals at δ 83.2 ppm and 77.0 ppm for C2 were assigned to 2,3-*trans* units catechin and afzelechin and 2,3-*cis* units epicatechin and epiafzelechin, respectively. On the other hand, the formation of polymeric structures is shown in the broad peaks at δ 107.4 ppm corresponding to the bonded monomer units C4→C8 and C4→C6, δ 73.6 ppm, which is consistent with the C3 extended units and δ 39–37 ppm corresponding to the C4 extended units.

In addition, the ^13^C-NMR spectra show signals characteristic of flavalignan-cinchonains (FL), as previously reported in the literature [21]. For instance, the signal at δ 168.7 ppm corresponding to the carbonyl C9″ from the lactone ring and the signals at δ 37.1 ppm and δ 34. 9 ppm are characteristic of the α-methylene (C8″) and β-methine (C7″), respectively (Figure 11). Further, the contrast between signals at δ 111.8 ppm and δ 105.1 ppm corresponding to C8 and C6, respectively, indicates substitution at C8, which is consistent with phenylpropanoid units [34].

### 2.3. Physicochemical Characterization of PA-E Loaded Nanoparticles

#### 2.3.1. Particle Size and Morphology

Morphology and size are critical to determining the stability and the loaded function of the nanoparticles due to the influence on the release of the compound inside the nanoparticle [35,36]. The obtained Z-average and PDI of nanoformulations F-1 and F-2 were used to determine their size and homogeneity in size. The results are summarized in Table 2; whereas HR-TEM mages in Figure 13 were used to assess their morphology.

The results showed a smaller particle size for the F-1 than for the F-2 formulation; however, both exhibited a similar PDI with a value around 0.4, indicating low aggregation and uniformity in the particle size [35,37]. On the other hand, both formulations showed quite similar dimensions observed by HR-TEM images and DLS data considering that these techniques are fundamentally different [38]. This parameter influences both the clearance and the biodistribution of nanoparticles due to nanoparticles of around 100 nm having a higher intracellular uptake [39], while large nanoparticles are easily taken up by the mononuclear phagocytic system [40].

Related to shape, HR-TEM images showed a similar spherical form for both formulations and F-1 exhibited as more open porous than F-2. This could be attributed to the amphiphilic block copolymer nature of PLU, which, according to its chain’s arrangement, could form hollows or more lipophilic sections [41].

#### 2.3.2. %EE and Encapsulation

The percentage entrapment efficiency (%EE) determines the amount of drug that has entered the carrier and it depends on combinatorial factors. It is an indicator of drug loading efficiency [42]. The %EE of the F-1 and F-2 formulations were 94% and 97%, respectively. It is important to analyze, with these factors, the high reproducibility and homogeneity of the nanoparticle [35].

FT-IR is an appropriate technique to confirm encapsulation [42,43]. The FT-IR spectra for formulations F-1 and F-2 are presented in Figure 14. By comparing the FT-IR spectra of unloaded formulations, the PA-E loaded nanoparticles F-1 and F-2, the free PA-E share a band at 3276 cm^−1^ due to the stretching of the -OH groups of alcohol and phenol compounds. The strong absorption at 2917 cm^−1^ can be assigned to –CH stretching vibrations in aliphatic and aromatic compounds. The signal at 1608 cm^−1^ indicated the fingerprint region of C=O. The band at 1056 cm^−1^ may be attributed to the C–O–C stretching mode of the aromatic ether linkage group [44].

Bands between 2800–3000 cm^−1^, which are characteristic of the asymmetric and symmetric stretching vibrations of CH_2_ and CH_3_ groups of the formulation components, were observed in the range of 2766–3019 cm^−1^ in F-1 and from 2752 to 3023 cm^−1^ F-2. The intense band, nearly at 3400 cm^−1^, is attributed to OH stretching, which corresponds to the lipid coating of synthesized NPs added to the interactions from the OH groups of the PA-E polyphenols. CHO exhibited one double band (C=C) on the second ring; this was prominently shown at 1640 cm^−1^. For both formulations, the results confirmed the presence of polymer, protein, surfactant, and the lipid coating on nanoparticles, also the PA-E. All the characteristic signals of the drug were present in the spectra of the formulations, indicating no chemical interaction between formulation components.

### 2.4. PA-E Loaded Nanoparticles In Vitro and In Vivo Evaluation

#### 2.4.1. In Vitro Release Evaluation of PA-E Loaded Nanoparticles and Dissolution Profile of Free PA-E

The selection of in vitro test dissolution media for nutraceuticals exhibiting limitate aqueous solubility becomes a challenge because of their hydrophobicity nature [45]. In this regard, some of the factors that need to be considered are the solubility of the drug in the medium and the ability of the medium to homogeneously solubilize the drug for accurate quantification of the released amount from a formulation at each time interval [45]. Therefore, surfactants and organic solvents in an appropriate concentration are allowed in dissolution media for poorly water-soluble drugs [46]. In addition, the selected medium should discriminate drug release patterns and help in the detection of variables in the formulation that could affect the drug release from the matrix [47].

Five dissolution media reported in the literature were tested preliminarily for both PA-E nanoformulations and free PA-E, considering the solubility of PA-E on them, the chemical instability of PA-E biomolecules at acidic or basic pHs [48] and their similarity with physiological conditions. HCl 0.1% (pH 3.2) [49], acetic acid 0.1% (pH 5.0) [50], phosphate-buffered saline containing 10% of EtOH (pH 7.2) [51], phosphate-buffered saline composed of 20% of MeOH and 2.5% of Tween (pH 7.4) [52] and water (pH 7.3) [53] were screened. Effectively, under acidic conditions, PA-E was not detected, and an abnormal dissolution profile of PA-E was observed in phosphate-buffered saline composed of 20% of MeOH and 2.5% of Tween (pH 7.4). Therefore, considering that the absorption pH in the lumen of the intestine is reported in the range of 6.8–7.4 [54] and the results obtained from the initial dissolution media evaluated, phosphate-buffered saline containing 10% of EtOH, pH 7.4 (M-1) was selected as a medium. Although water is not recommended as a release medium for poorly water-soluble molecules, it was also tested in an attempt to evaluate the aqueous solubility enhancement exerted by the synthesized nanosystems.

The in vitro release profiles of PA-E from the hybrid nanoparticles F-1 and F-2 as well as the dissolution profiles of free PA-E in two dissolution media are shown in Figure 15. The release of PA-E from F-1 and F-2 showed as erratic and similar for both formulations. In addition, no significant improvement was observed in relation to the free PA-E profile. Nevertheless, comparing the dissolution profile of free PA-E with PA-E released from both nanoformulations in water (M-2) showed differences in the release of PA-E in favor of F-1. At 120 min, only 5% of the free PA-E was dissolved while its release from the F-1 was around 70% and F-2 remained close to 25%. The results suggest that M-1 was not able to discriminate between different formulations.

Diverse factors contributed to an efficient release; some of them are the large surface area, a high diffusion coefficient due to small molecular size, low viscosity in the matrix and a short diffusion distance δ for the drug (i.e., release from the outer surface region of the nanoparticle). The increase in release rate with decreasing particle size was described for drugs incorporated in a matrix with a polymer or a protein [18] in agreement with the results presented herein. F-1 exhibited a smaller size and higher release rate than F-2. The difference in the release profile in favor of F-1 can also be explained due to the degradation of the PLU backbone in alignment with some studies that have reported the degradation of the PLU backbone in water, which could contribute to water solubilization and subsequent release of the drug [55,56,57].

#### 2.4.2. Antioxidant Activity Evaluation of Free and Nanoencapsulated PA-E from *U. tomentosa*

Antioxidant activity of free and nanoencapsulated PA-E was studied through DPPH analysis, as described in the experimental section. Results of the antioxidant activity are summarized in Table 3.

Evaluation of the antioxidant activity of free PA-E samples prepared in EtOH and water was performed through a DPPH assay. The antioxidant activity found for PA-E aligns with previous results, reporting procyanidins and propelargonidins, which are its main constituents, indicating these compounds as responsible for antioxidant activity [8]. One-way ANOVA followed by Tukey’s post hoc test shows a significant difference (*p* < 0.05) between samples of free PA-E when prepared in ethanol or water. Results for free PA-E in EtOH presented a significantly lower IC_50_, therefore higher antioxidant activity, than free PA-E in aqueous solution, evidencing the lower solubility of the PA-E in water.

The DPPH assay of PA-E nanoparticles showed an important decrease in the IC_50_, hence, a higher antioxidant activity than free PA-E in an aqueous solution. IC_50_ of F-1 nanoparticles was shown to be similar to that of free PA-E in ethanolic solution, while F-2 nanoparticles improved the antioxidant activity by 8%. Results show that the antioxidant activity was significantly enhanced by nanoencapsulation for both formulations tested. These observations are consistent with previous results obtained on nanoparticle formulations of proanthocyanidins [58,59], where their antioxidant activity was enhanced in nanoformulations, and constitute, to our knowledge, the first report for nanoencapsulated proanthocyanidins from *U. tomentosa*.

#### 2.4.3. Evaluation of the Cellular Immune Response of Mice through DTH

The delayed-type hypersensitivity (DTH) reaction for immune response is a technically simple test capable of reflecting the development of systemic antigen-specific immunity when exposing the subject to different treatments. The data presented here demonstrate that antigen-specific DTH responses are influenced by the daily consumption of flavan-3-ols, with nanoformulations being the samples with the greatest responsiveness at 24 and 48 h, as shown in Figure 16.

When evaluating the different samples, we observed that nanoformulations F-1 and F-2 of *U. tomentosa* PA-E were the two with the greatest effectiveness and influence on the immune response in the treated mice, who showed increased response in the treatment when compared to the negative control in group I. This differential activity was detected at 48 h after the application of the antigen, according to the post hoc comparisons (Table 4). At 24 h, the immune response was not different among samples after FLH antigen application and the control that did not receive the FLH antigen did not show any immune response (Figure 16).

*U. tomentosa* leaves proanthocyanidin extract PA-E was found to also exert differential activity when compared with the control, but less so than both nanoformulations. In fact, *U. tomentosa* extracts, as mentioned, are known for their immune stimulation properties, but were previously attributed almost exclusively to oxindole alkaloids. In turn, our samples are constituted of proanthocyanidins in our enriched PA-E, whereas these results align with previous reports on proanthocyanidins from other substrates [60] ameliorating immune suppression in mice.

The DTH reaction constitutes an important in vivo display of a biological response mediated by the degree of recruitment of both CD4+ and CD8+ lymphocytes to the mouse pad [61,62]. This reaction is observed during the first 24 to 48 h after the antigen inoculation, showing to measure the defense response against intracellular infectious agents [63]. DTH is an indicator of cancer prognosis and improvement when administering tumor antigens to patients [64]. Previously reported data have also demonstrated that tumor antigen-specific DTH responses correlate significantly to a measurable antigen-specific peripheral blood T-cell response [65].

Recent therapeutic approaches showed that the clinical response rate is very low for modern approaches for boosting the immune response, such as vaccination with tumor antigens [66]. Within this scenario, endogenous immunity to release a strong antitumor response has significant advantages over conventional and modern cancer therapies [67]. The enhancement of the immune response by external therapeutic agents is then required since the immune system often fails to control the spread of cancerous malignancies and other immune-based diseases.

On the other hand, the ability of plant metabolites to boost cell-mediated immune responses has been reported for different plant products [68,69,70]. Other studies in the fish model demonstrated dietary supplementation of *U. tomentosa* enhanced the immune activity and growth performance [71]. In this case, the most important bioactive compounds found in this plant were oxindole alkaloids, derivatives of quinovic acid glycosides, and low molecular weight polyphenols [72], while our PA-E is composed of proanthocyanidins. Therefore, results on this extract and its nanoformulations F-1 and F-2, constitute to our knowledge, the first report on this DTH activity due to proanthocyanidins present in *U. tomentosa* leaves and further, according to our findings, the immune response to the antigen is augmented when these compounds are nanoformulated in F-1 and F-2 samples.

Therefore, our murine model set the foundations for further research on the discovery of the potent immune stimulation of the flavan-3-ols with higher molecular weight from *U. tomentosa* that, administered as a nanoformulation can improve its effects, resulting in augmented antigen-specific responses. Since immune cellular responses of CD8+ T cells have remarkable diverse roles in health, as is demonstrated by being involved in autoimmunity, chronic infection and cancer [72,73], our findings suggest that hybrid nanoformulations from *U. tomentosa* could have implications for regulating the immune response in such.

## 3. Materials and Methods

### 3.1. Materials and Reagents

Cholesterol (CHO), Bovine Serum Albumin (BSA, lyophilized powder, ≥96%), Pluronic^®^ F-127 (PLU), 2,2-diphenyl-1-picrylhidrazyl (DPPH), phosphoric acid (H_3_PO_4_), Methanol-D4 (CD_3_OD), disodium hydrogen phosphate and LC-Mass spectroscopy water were purchased from Sigma–Aldrich (Burlington, MA, USA). Sodium dihydrogen phosphate monohydrate and formic acid were acquired from Merck (Rahway, NJ, USA). Polysorbate 80 (Tween 80^®^) was purchased from Sonntag & Rote S.A. (San José, Costa Rica), and sorbitan monooleate (Span^®^ 80) was supplied by LABQUIMAR S.A. (San José, Costa Rica). Chloroform (CHCl_3_), methanol (MeOH), acetonitrile (MeCN), methyl tert-butyl ether (MTBE) and hexane were purchased from JTBaker. All solvents were highly pure or of HPLC/UV grade, and the water was purified using a Millipore system filtered through a Millipore membrane 0.22 µm Millipak 40 filter.

### 3.2. Proanthocyanidins Extract from U. tomentosa Leaves

*U. tomentosa* leaves were acquired from the northern region of Costa Rica and the voucher was deposited in the Costa Rican National Herbarium. The leaves were dried on a stove at 40 °C, being turned over every 24 h for a week until totally dry. Extraction was performed as described before [4]. Briefly, the dried leaves material was extracted in a mixture of MTBE and MeOH 90:10 (*v*/*v*) at 25 °C during 30 min in ultrasound; subsequently, it was left standing for 24 h to obtain a non-polar extract of the samples. The solvent was removed by filtration and the extraction process was repeated once. The extracts were concentrated to dryness and washed with MeOH to extract residual polyphenols. After the non-polar extraction, the residual material was extracted with MeOH at 25 °C during 30 min in ultrasound and then it was left standing for 24 h. The solvent was removed by filtration and the extraction process was repeated twice. These three MeOH extracts were combined with the MeOH washings of the dried non-polar extracts. MeOH was evaporated in a rotavapor at less than 40 °C, and the dried extract was then washed with hexane, MTBE and CHCl_3_ successively to deliver the extract (PA-E).

### 3.3. UPLC-QTOF-ESI-MS Analysis

The UPLC-MS system used to analyze the composition of *U. tomentosa* leaves PA-E consisted of a Xevo G2-XS QTOF (Waters, UK) coupled with an AQUITY H Class UPLC system with a quaternary pump. The ESI source parameters were set to a capillary voltage of 2 kV, a sampling cone of 20 eV, a source temperature of 150 °C, and a source offset of 10 °C. The desolvation temperature was set to 450 °C, the cone gas flow to 0 L/h, and the desolvation gas flow to 900 L/h.

The measurement was performed in an MS^e^ high-resolution negative mode using an acquisition mass range from 100 *m*/*z* to 1000 *m*/*z* and a scan rate of 0.5 s, where fragmentation was carried out using independent data acquisition for all eluting compounds with a collision energy ramp from 20 V to 30 V, storing at the high-energy function. Instrument calibration was applied in the mass range of the measurement with sodium formate. Lock mass correction was applied directly to the measurement using a leucine enkephalin infusion measured every 30 s during the run. The data was analyzed using the MassLynx V4.2 software from Waters. 

The separation was carried out on a Luna RP-C18 column (150 mm × 4.6 mm i.d. × 4 µm, Phenomenex, Torrance, CA, USA) with a pre-column filter (Phenomenex, Torrance, CA, USA). In total, 1 uL of the sample was injected at a flow rate of 0.5 mL/min using a chromatographic gradient starting at 87%A, 8% B and 5%C, changing to 48% A, 40%B and 12%C at 35 min, then to 0%A, 85%B, and 15%C at 40 min, holding it for 6 min, and then the column was equilibrated for 10 min to initial conditions. Solvents used in the mobile phase were A water with 0,1% formic acid, B MeOH with 0,1% formic acid, and C MeCN with 0,1% formic acid.

### 3.4. NMR ^13^C Analysis of U. tomentosa Polyphenolic Extract

^13^C-NMR analysis was performed using 10 mg of *U. tomentosa* polyphenolic extract PA-E and 0.5 mL of CD_3_OD. Spectra were recorded on a Bruker Ascend 400 MHz instrument and chemical shifts are reported in ppm relative to internal tetramethylsilane (TMS, δ 0.0 ppm) as standard.

### 3.5. Preparation and Characterization of PA-E Loaded Nanoparticles

PA-E loaded hybrid polymeric-lipid nanoparticles (F-2) were prepared accordingly with Wilhelm et al. (2021) [74]. Meanwhile, PA-E loaded hybrid protein-lipid nanoparticles (F-2) were prepared based on the aforementioned method using BSA instead of PLU. In brief, 5 mg of PA-E was dissolved in 6 mL of MeOH: CHCl_3_ 1:1 mixture along with 120 mg of CHO. Separately, an aqueous phase was prepared using 250 mg of PLU for F-1 or 250 mg of BSA for F-2 in 50 mL of acetic acid 0.1% and adding a 1:1 mixture of Tween 80^®^: Span80^®^ 1:1. Then, the aqueous solution was added to the organic one and homogenized at 16,000 rpm for 10 min. The nanoparticles were collected by ultracentrifugation using a Thermo Scientific Sorvall ST 16R centrifuge at 12,000 rpm for 40 min at 10 °C. To remove the remaining unencapsulated substrate and unreacted substances, the nanoparticles were washed three times with ultrapure water. The final formulation was dispersed in 5 mL of purified water containing 0.01% Tween80^®^; filtered through an ADVANTEC^®^ (Tokyo, Japan) ultrafilter unit and refrigerated up to further characterization. Blanks of F-1 and F-2 were prepared as mentioned above without adding PA-E in the organic phase.

#### 3.5.1. High-Resolution Transmission Electron Microscopy (HR-TEM)

PA-E loaded nanoparticles were evaluated using a JEOL, JEM2011 HR-TEM at an acceleration voltage of 120 kV. Samples were prepared by placing 5 uL of each suspension and drying under a nitrogen atmosphere.

#### 3.5.2. Dynamic Light Scattering (DLS)

The particle size (PS, z-average) and polydispersity index (PI) were measured using a Malvern Nano Zetasizer ZS90 (Malvern Panalytical, Malvern, UK) instrument using a medium refractive index of 1.33, and viscosity of 0.8872 cP under 90°. Samples were diluted with deionized water to achieve appropriate concentrations, and measurements were done at 25 °C.

#### 3.5.3. Total Attenuate Fourier Transform Infrared Spectroscopy (ATR-FT-IR)

The ATR-FT-IR spectra of the sample were recorded on a Thermo Scientific Nicolet 6700 FT-IR spectrometer. The samples were placed directly into the ATR diamond cell without further preparation and analyzed in the range of 4000–600 cm^−1^.

#### 3.5.4. Encapsulation Efficiency (EE)

The %EE was calculated through the indirect method of quantifying the PA-E remaining in the supernatant of F-1 and F-2 after ultracentrifugation and applying Equation (1). The concentration of PA-E in the solution was determined using a Shimadzu 1800 double beam UV-Vis spectrophotometer at a wavelength of 273 nm.
(1)$$\%\mathrm{EE}=\frac{\left[\mathrm{Total}\;\mathrm{PA}-E\;\mathrm{content}\;\left(\mathrm{mg}\right)\right]-\left[\mathrm{PA}-E\;\mathrm{content}\;\mathrm{in}\;\mathrm{supernatant}\;\left(\mathrm{mg}\right)\right]\;}{\mathrm{Total}\;\mathrm{PA}-E\;\mathrm{content}\;\left(\mathrm{mg}\right)}\times 100\;$$

#### 3.5.5. In Vitro PA-E Release Evaluation

The PA-E release profile from formulations F-1 and F-2 as well as the dissolution profile of free PA-E were estimated using two different dissolution media. The first one is composed of phosphate-buffered saline of pH 7.4 containing 10% of EtOH (M-1) and water (M-2). A total of 1 mL of F-1 or F-2 was immersed in 80 mL of the respective medium and maintained at 37 ± 0.5 °C and 150 rpm in a Labnet 211 DS shaking incubator. A total of 4 mL of each solution was withdrawn at specific time intervals without replacing the volume. The aliquots were centrifuged at 6000 rpm for 10 min in a Thermo Scientific Sorvall ST 16R centrifuge maintained at 37 °C. The concentration of PA-E in the solutions was measured using the UV-Vis method indicated in Section 3.5.4. The sampling was done in triplicate.

#### 3.5.6. DPPH Radical-Scavenging Activity

The DPPH evaluation of antioxidant activity was performed as previously reported [8], for free PA-E and formulations F-1 and F-2. Free PA-E samples for antioxidant evaluation were prepared either in an ethanolic or aqueous solution. Samples of nanoformulations F-1 and F-2 containing PA-E were prepared in water. Briefly, a solution of 2,2-diphenyl-1-picrylhydrazyl (DPPH) (0.25 mM) was prepared using ethanol as the solvent. Next, 0.5 mL of this solution was mixed with 1 mL of the respective free or nanoencapsulated PA-E solution at different concentrations and incubated at 25 °C in the dark for 30 min. DPPH absorbance was measured at 517 nm. Blanks were prepared for each concentration. The percentage of the radical-scavenging activity of the sample was plotted against its concentration to calculate IC_50_, which is the amount of sample required to reach the 50% radical-scavenging activity. The controls were prepared for each assay using solvent instead of samples. The percentage of the radical-scavenging activity inhibition was calculated for each concentration according to Equation (2). This percentage was plotted against the sample concentrations to calculate IC_50_, which is the amount required to reach 50% inhibition of DPPH radical-scavenging activity. The samples were analyzed this way in three independent assays.
(2)$$\mathrm{Inhibition}\;\mathrm{percentage}=\frac{\left(\mathrm{Absorbance}\;\mathrm{of}\;\mathrm{control}\;-\;\mathrm{Absorbance}\;\mathrm{of}\;\mathrm{Trolox}\;\mathrm{or}\;\mathrm{sample}\right)}{\left(\mathrm{Absorbance}\;\mathrm{of}\;\mathrm{control}\right)}\times 100$$

#### 3.5.7. Evaluation of the Cellular Immune Response of Mice through the Delayed-Type Hypersensitivity

Animals reared in the Laboratorio de Ensayos Biológicos (LEBi) of the Universidad de Costa Rica were used. The assay was performed at the bioterium of the Faculty of Pharmacy (conditions: constant temperature of 22 to 24 °C, relative humidity of 60% to 70%, and light and dark cycles of 12 h each). The animals had access to water and food ad libitum. Their handling and care were approved by the CICUA-023-2020 permit from the Institutional Committee for the Care and Use of Animals (CICUA, for its Spanish Acronym) at session No. 200–2020.

Briefly, two to four-month-old female mice of the C3H/He strain were separated into groups of three animals and individualized in separate boxes. Subsequently, they were orally administered water (groups 1 and 2) or 40 mg of one of the treatments of interest (PA-E for group 3, F-1 for group 4, and F-2 for group 5). The administration took place from Monday to Friday mornings for five weeks (day 1 to day 36), starting on Monday. Group 1 was used as a control without the application of the *Fissurella latimarginata* hemocyanin (FLH) antigen. Animals in groups 2 to 6 were immunized with FLH (50 µg) subcutaneously on days 7 and 21.

The DTH reaction was performed according to previous studies [75]. Within each group of three animals, the thickness of the pad of the right and left feet was measured using a high-precision instrument Krœplin C1R10 (Krœplin GmbH), both before the injection of 50 µg of FLH (day 39) and at 24, 48 and 72 h (days 40, 41 and 41, respectively). The DTH reaction was graphed as the difference in thickness between the left and the right foot.

### 3.6. Statistical Analysis

One-way analysis of variance (ANOVA) with Tukey’s post hoc as statistical tests were applied to determine differences in DPPH antioxidant activity and in quantitative values with respect to the footpad thickness within each hour of measure in the DTH reaction.

## 4. Conclusions

This paper reports valuable information about the phenolic composition of *U. tomentosa* leaves and constitutes a detailed report of the characterization of a PA-E from *U. tomentosa* leaves from Costa Rica, using UPLC-QTOF-ESI MS for a total of 50 compounds, including: 2 monomers, epicatechin and catechin, 11 procyanidin oligomers from (epi)catechin, 10 propelargonidin oligomers from (epi)afzelechin, 7 flavalignans and 20 mixed propelargonidin dimers and trimers from (epi)catechin and (epi)afzelechin units, constituting for the first time that propelargonidin trimers are reported for *U. tomentosa* leaves. Further, PA-E was successfully loaded into two different hybrid nanoparticles, F-1 and F-2 composed of PLU and BSA, respectively. The nanoformulations presented appropriate characteristics of particle size, monodisperse size distribution, spherical shape, and highly efficient encapsulation. Both formulations exhibited considerable improvements in vitro antioxidant activity and in vivo; the delayed-type hypersensitive test shows that both hybrid nanoparticles can improve their effects, resulting in increased antigen-specific responses. The results constitute, to our knowledge, the first report on this DTH activity due to proanthocyanidins present in *U. tomentosa* leaves and its augmented immune response to the antigen when these compounds were administrated in nanosystems. Our findings suggest that hybrid nanoformulations from *U. tomentosa* proanthocyanidins could have implications in regulating immune cellular response in diseases, such as autoimmunity, chronic infection and cancer.

## Figures and Tables

**Figure 1 plants-11-01737-f001:**
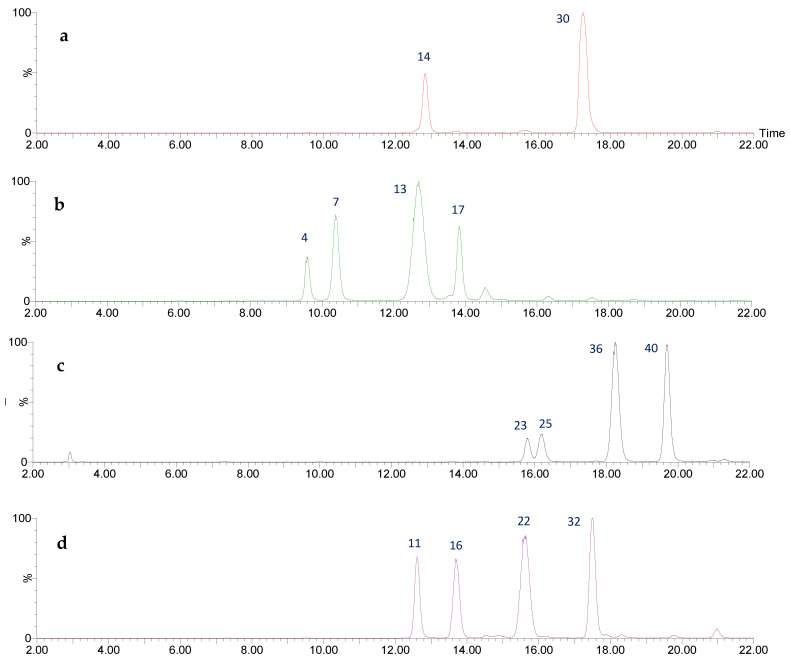
UPLC-QTOF-ESI MS chromatograms (from top to bottom) of: (**a**) flavan-3-ol monomers, (**b**) procyanidin (epi)catechin dimers (**c**) propelargonidin (epi)afzelechin-(epi)afzelechin dimers, (**d**) propelargonidin (epi)afzelechin-(epi)catechin dimers.

**Figure 2 plants-11-01737-f002:**
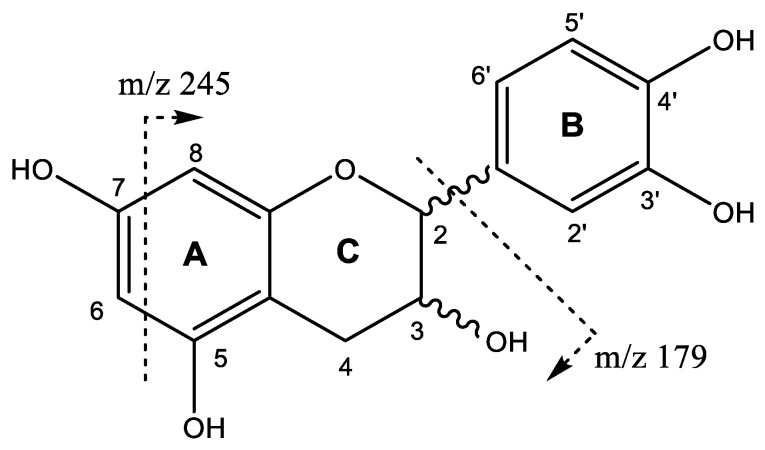
Flavan-3-ol monomers structure and main fragments.

**Figure 3 plants-11-01737-f003:**
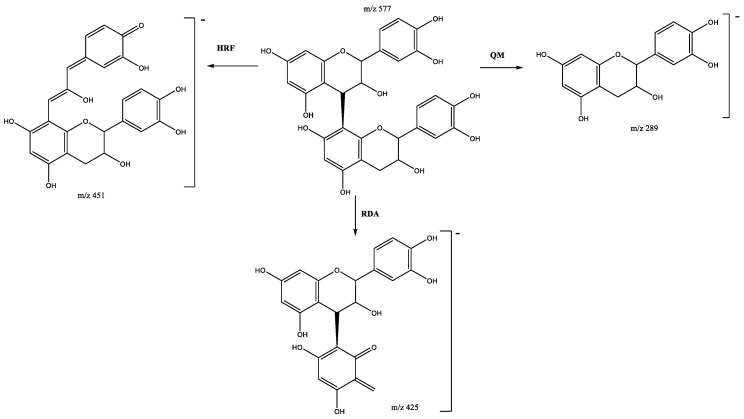
Fragmentation pathway of B-type procyanidin dimer showing the products formed by heterocyclic ring fusion (HRF), quinone methide (QM) and retro-Diels–Alder (RDA) reactions.

**Figure 4 plants-11-01737-f004:**
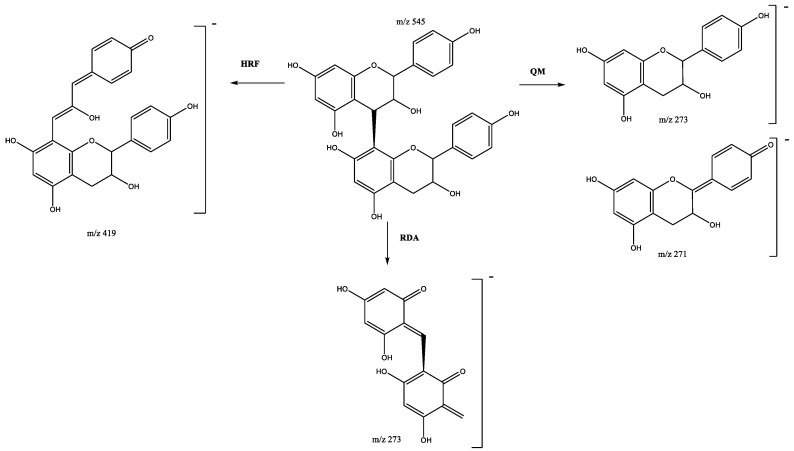
Propelargonidin (epi)afzelechin dimers fragmentation pathway.

**Figure 5 plants-11-01737-f005:**
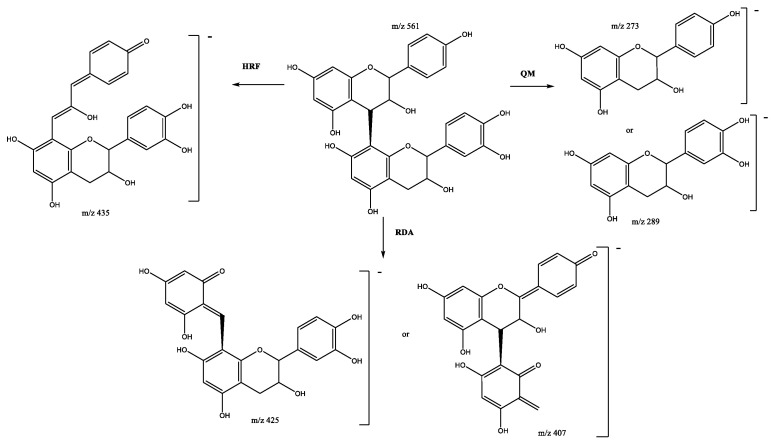
Propelargonidin (epi)afzelechin-(epi)catechin B-dimer fragmentation pathway.

**Figure 6 plants-11-01737-f006:**
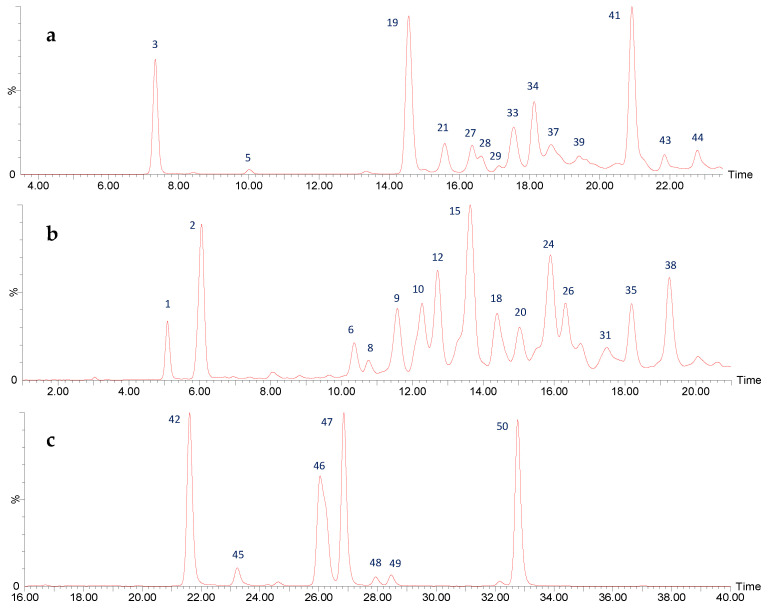
UPLC-QTOF-ESI MS chromatograms of (**a**) properlagonidin (epi)afzelechin-(epi)afzelechin-(epi)afzelechin and (epi)afzelechin-(epi)afzelechin-(epi)catechin B-trimers, (**b**) properlagonidin (epi)afzelechin-(epi)catechin-(epi)catechin B-trimers and procyanidin (epi)catechin-(epi)catechin-(epi)catechin B-trimers, and (**c**) flavalignans.

**Figure 7 plants-11-01737-f007:**
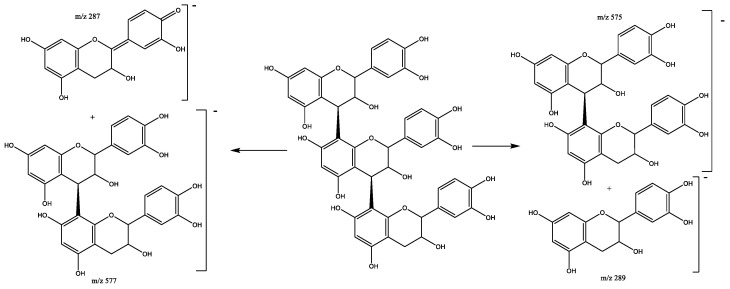
Procyanidin B-trimer fragmentation pathway.

**Figure 8 plants-11-01737-f008:**
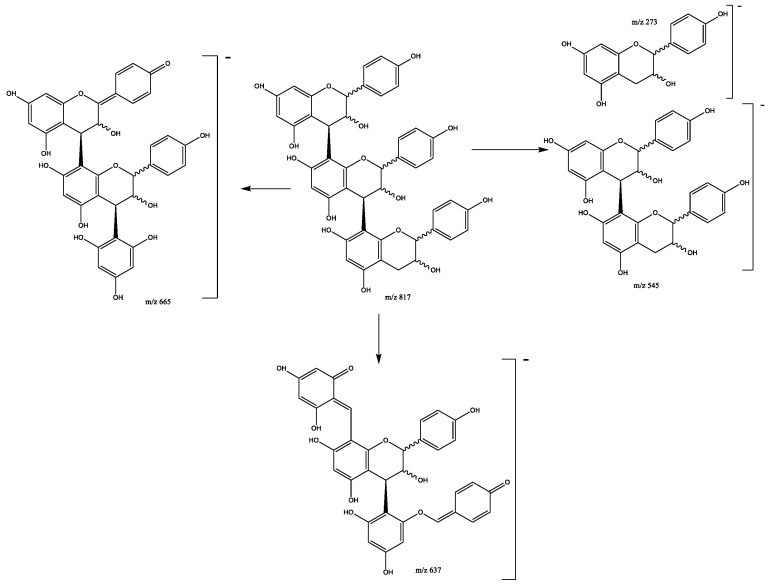
Propelargonidin (epi)afzelechin-(epi)afzelechin-(epi)afzelechin B-trimer fragmentation pattern.

**Figure 9 plants-11-01737-f009:**
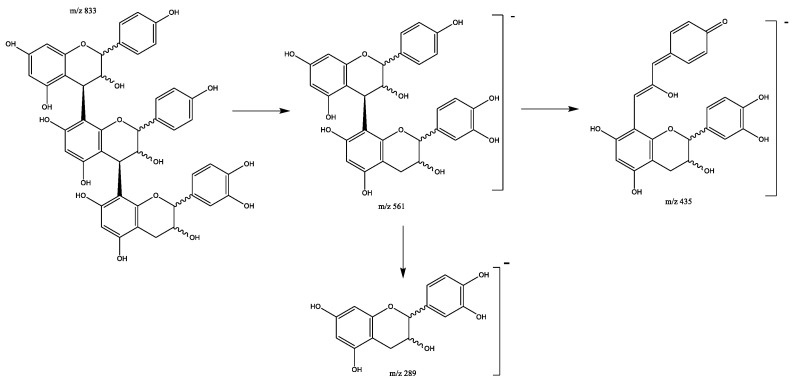
Propelargonidins (epi)afzelechin-(epi)afzelechin-(epi)catechin B-trimers fragmentation pattern.

**Figure 10 plants-11-01737-f010:**
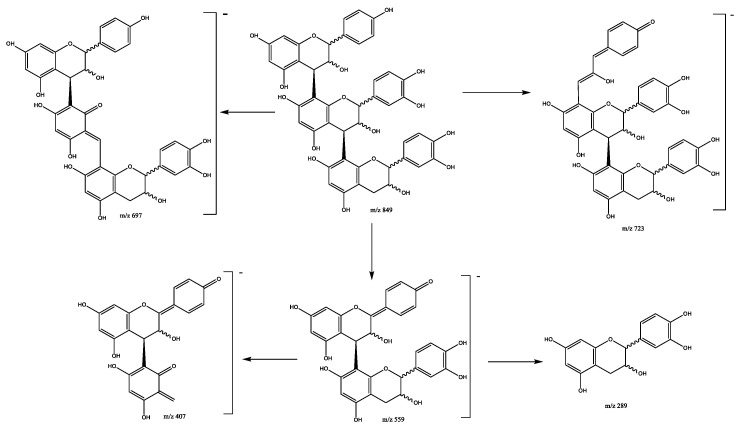
Propelargonidins (epi)afzelechin-(epi)catechin-(epi)catechin B-trimers fragmentation pathway.

**Figure 11 plants-11-01737-f011:**
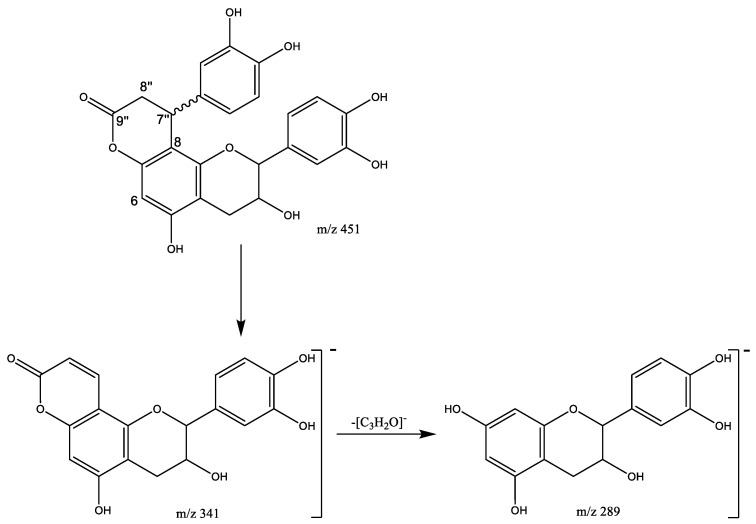
Cinchonains fragmentation pathway.

**Figure 12 plants-11-01737-f012:**
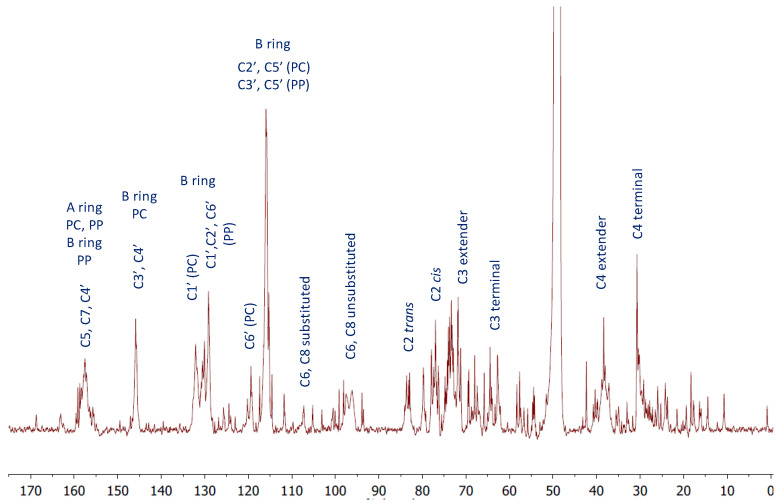
^13^C-NMR (MeOD) for a polyphenolic fraction of leaves (LH-F3) from *U. tomentosa*.

**Figure 13 plants-11-01737-f013:**
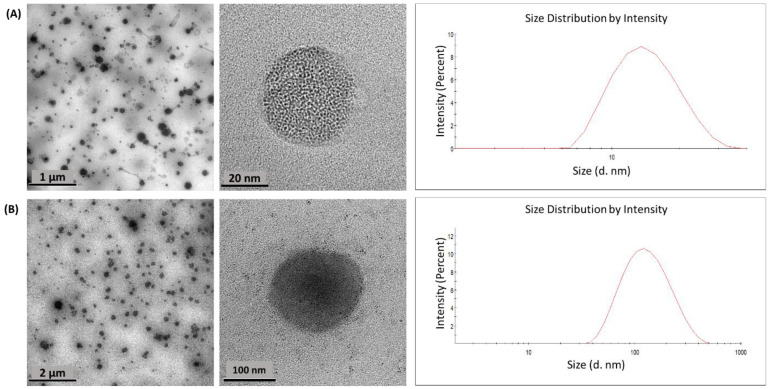
HR-TEM micrographs and histogram size distributions of (**A**) F-1 and (**B**) F-2.

**Figure 14 plants-11-01737-f014:**
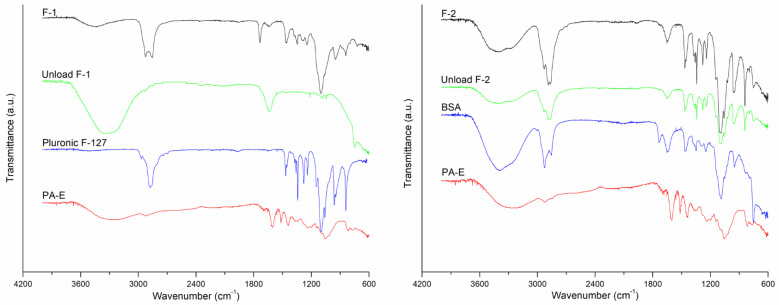
FT-IR spectra of F-1 and F-2 formulations.

**Figure 15 plants-11-01737-f015:**
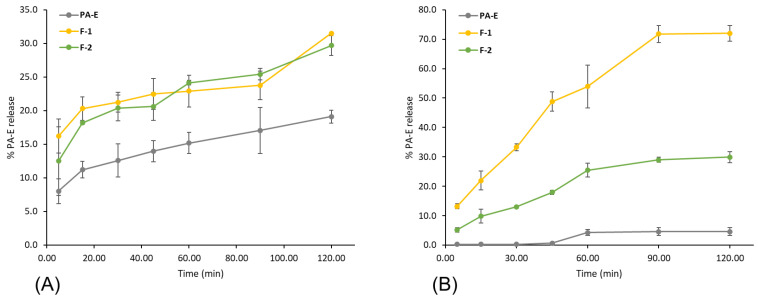
Release profiles of F-1 and F-2 compared to free PA-E dissolution profile in two dissolution media: (**A**) M-1 and (**B**) M-2. Error bars represent the standard deviation of PA-E concentration in the three independent samplings.

**Figure 16 plants-11-01737-f016:**
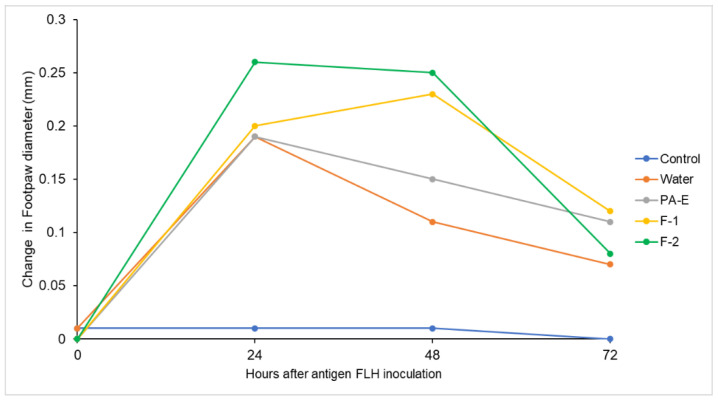
Delayed-type hypersensitivity (DTH) test expressed as change in footpad thickness for each treatment measured at 0, 24, 48 and 72 h.

**Table 1 plants-11-01737-t001:** Profile of proanthocyanidins and flavalignans identified by UPLC-QTOF-ESI MS analysis of *U. tomentosa* leaves.

No.	Rt (min)	Tentative Identification	Formula	[M-H]^−^	Error (ppm)	MS2 Fragments
1	5.09	Procynidin B trimer [(epi)cat-(epi)cat-(epi)cat] (I of VII)	C_45_H_37_O_18_	865.2084	−4.966	577, 575, 425, 407, 289, 287, 175, 137
2	6.04	Propelargonidin trimer [(epi)afz-(epi)cat-(epi)cat] (I of VIII)	C_45_H_37_O_17_	849.2089	7.506	723, 697, 679, 561, 559, 407, 289
3	7.33	Properlargonidin trimer [(epi)afz-(epi)afz-(epi)cat]] (I of VIII)	C_45_H_37_O_16_	833.2141	2.356	711, 707, 561, 543, 435, 289, 273, 271, 174
4	9.58	Pocyanidin B dimer [(epi)cat-(epi)cat] (I of IV)	C_30_H_25_O_12_	577.1382	7.186	559, 451, 435, 425, 407, 289
5	10.01	Propelargonidin trimer [(epi)afz-(epi)afz-(epi)afz] (I of VI)	C_45_H_37_O_15_	817.2179	6.367	779, 775, 665, 637, 578, 527, 273, 237
6	10.35	Procynidin B trimer [(epi)cat-(epi)cat-(epi)cat] (II of VII)	C_45_H_37_O_18_	865.2084	−4.966	577, 575, 425, 407, 289, 287, 175, 137
7	10.38	Pocyanidin B dimer [(epi)cat-(epi)cat] (II of IV)	C_30_H_25_O_12_	577.1382	7.186	559, 451, 435, 425, 407, 289
8	10.77	Procynidin B trimer [(epi)cat-(epi)cat-(epi)cat] (III of VII)	C_45_H_37_O_18_	865.2084	−4.966	577, 575, 425, 407, 289, 287, 175, 137
9	11.55	Procynidin B trimer [(epi)cat-(epi)cat-(epi)cat] (IV of VII)	C_45_H_37_O_18_	865.2084	−4.966	577, 575, 425, 407, 289, 287, 175, 137
10	12.29	Propelargonidin trimer [(epi)afz-(epi)cat-(epi)cat] (II of VIII)	C_45_H_37_O_17_	849.2089	7.506	723, 697, 679, 561, 559, 407, 289
11	12.62	Propelargonidin dimer [(epi)afz-(epi)cat] (I of IV)	C_30_H_25_O_11_	561.1416	4.388	435, 425, 407, 289, 273, 271, 245
12	12.67	Procynidin B trimer [(epi)cat-(epi)cat-(epi)cat] (V of VII)	C_45_H_37_O_18_	865.2084	−4.966	577, 575, 425, 407, 289, 287, 175, 137
13	12.70	Pocyanidin B dimer [(epi)cat-(epi)cat] (III of IV)	C_30_H_25_O_12_	577.1382	7.186	559, 451, 435, 425, 407, 289
14	12.83	Catechin	C_15_H_13_O_6_	289.0729	7.733	245, 179
15	13.63	Propelargonidin trimer [(epi)afz-(epi)cat-(epi)cat] (III of VIII)	C_45_H_37_O_17_	849.2089	7.506	723, 697, 679, 561, 559, 407, 289
16	13.68	Propelargonidin dimer [(epi)afz-(epi)cat] (II of IV)	C_30_H_25_O_11_	561.1416	4.388	435, 425, 407, 289, 273, 271, 245
17	13.84	Pocyanidin B dimer [(epi)cat-(epi)cat] (IV of IV)	C_30_H_25_O_12_	577.1382	7.186	559, 451, 435, 425, 407, 289
18	14.39	Procynidin B trimer [(epi)cat-(epi)cat-(epi)cat] (VI of VII)	C_45_H_37_O_18_	865.2084	−4.966	577, 575, 425, 407, 289, 287, 175, 137
19	14.56	Properlargonidin trimer [(epi)afz-(epi)afz-(epi)cat]] (II of VIII)	C_45_H_37_O_16_	833.2141	2.356	711, 707, 561, 543, 435, 289, 273, 271, 174
20	15.02	Propelargonidin trimer [(epi)afz-(epi)cat-(epi)cat] (IV of VIII)	C_45_H_37_O_17_	849.2089	7.506	723, 697, 679, 561, 559, 407, 289
21	15.57	Properlargonidin trimer [(epi)afz-(epi)afz-(epi)cat]] (III of VIII)	C_45_H_37_O_16_	833.2141	2.356	711, 707, 561, 543, 435, 289, 273, 271, 174
22	15.63	Propelargonidin dimer [(epi)afz-(epi)cat] (III of IV)	C_30_H_25_O_11_	561.1416	4.388	435, 425, 407, 289, 273, 271, 245
23	15.79	Propelargonidin dimer [(epi)afz-(epi)afz] (I of IV)	C_30_H_25_O_10_	545.1473	5.644	419, 409, 287, 273, 271
24	15.89	Propelargonidin trimer [(epi)afz-(epi)cat-(epi)cat] (V of VIII)	C_45_H_37_O_17_	849.2089	7.506	723, 697, 679, 561, 559, 407, 289
25	16.18	Propelargonidin dimer [(epi)afz-(epi)afz] (II of IV)	C_30_H_25_O_10_	545.1473	5.644	419, 409, 287, 273, 271
26	16.30	Procynidin B trimer [(epi)cat-(epi)cat-(epi)cat] (VII of VII)	C_45_H_37_O_18_	865.2084	−4.966	577, 575, 425, 407, 289, 287, 175, 137
27	16.36	Properlargonidin trimer [(epi)afz-(epi)afz-(epi)cat]] (IV of VIII)	C_45_H_37_O_16_	833.2141	2.356	711, 707, 561, 543, 435, 289, 273, 271, 174
28	16.58	Properlargonidin trimer [(epi)afz-(epi)afz-(epi)cat]] (V of VIII)	C_45_H_37_O_16_	833.2141	2.356	711, 707, 561, 543, 435, 289, 273, 271, 174
29	17.15	Propelargonidin trimer [(epi)afz-(epi)afz-(epi)afz] (II of VI)	C_45_H_37_O_15_	817.2179	6.367	779, 775, 665, 637, 578, 527, 273, 237
30	17.24	Epicatechin	C_15_H_13_O_6_	289.0729	7.733	245, 179
31	17.45	Propelargonidin trimer [(epi)afz-(epi)cat-(epi)cat] (VI of VIII)	C_45_H_37_O_17_	849.2089	7.506	723, 697, 679, 561, 559, 407, 289
32	17.50	Propelargonidin dimer [(epi)afz-(epi)cat] (IV of IV)	C_30_H_25_O_11_	561.1416	4.388	435, 425, 407, 289, 273, 271, 245
33	17.56	Properlargonidin trimer [(epi)afz-(epi)afz-(epi)cat]] (VI of VIII)	C_45_H_37_O_16_	833.2141	2.356	711, 707, 561, 543, 435, 289, 273, 271, 174
34	18.13	Properlargonidin trimer [(epi)afz-(epi)afz-(epi)cat]] (VII of VIII)	C_45_H_37_O_16_	833.2141	2.356	711, 707, 561, 543, 435, 289, 273, 271, 174
35	18.21	Propelargonidin trimer [(epi)afz-(epi)cat-(epi)cat] (VII of VIII)	C_45_H_37_O_17_	849.2089	7.506	723, 697, 679, 561, 559, 407, 289
36	18.24	Propelargonidin dimer [(epi)afz-(epi)afz] (III of IV)	C_30_H_25_O_10_	545.1473	5.644	419, 409, 287, 273, 271
37	18.57	Propelargonidin trimer [(epi)afz-(epi)afz-(epi)afz] (III of VI)	C_45_H_37_O_15_	817.2179	6.367	779, 775, 665, 637, 578, 527, 273, 237
38	19.27	Propelargonidin trimer [(epi)afz-(epi)cat-(epi)cat] (VIII of VIII)	C_45_H_37_O_17_	849.2089	7.506	723, 697, 679, 561, 559, 407, 289
39	19.42	Propelargonidin trimer [(epi)afz-(epi)afz-(epi)afz] (IV of VI)	C_45_H_37_O_15_	817.2179	6.367	779, 775, 665, 637, 578, 527, 273, 237
40	19.71	Propelargonidin dimer [(epi)afz-(epi)afz] (IV of IV)	C_30_H_25_O_10_	545.1473	5.644	419, 409, 287, 273, 271
41	20.91	Properlargonidin trimer [(epi)afz-(epi)afz-(epi)cat]] (VIII of VIII)	C_45_H_37_O_16_	833.2141	2.356	711, 707, 561, 543, 435, 289, 273, 271, 174
42	21.61	Cinchonain (I of VII)	C_24_H_19_O_9_	451.1039	−1.915	341, 289, 271
43	21.84	Propelargonidin trimer [(epi)afz-(epi)afz-(epi)afz] (V of VI)	C_45_H_37_O_15_	817.2179	6.367	779, 775, 665, 637, 578, 527, 273, 237
44	22.79	Propelargonidin trimer [(epi)afz-(epi)afz-(epi)afz] (VI of VI)	C_45_H_37_O_15_	817.2179	6.367	779, 775, 665, 637, 578, 527, 273, 237
45	23.22	Cinchonain (II of VII)	C_24_H_19_O_9_	451.1039	−1.915	341, 289, 271
46	26.06	Cinchonain (III of VII)	C_24_H_19_O_9_	451.1039	−1.915	341, 289, 271
47	26.85	Cinchonain (IV of VII)	C_24_H_19_O_9_	451.1039	−1.915	341, 289, 271
48	27.94	Cinchonain (V of VII)	C_24_H_19_O_9_	451.1039	−1.915	341, 289, 271
49	28.47	Cinchonain (VI of VII)	C_24_H_19_O_9_	451.1039	−1.915	341, 289, 271
50	32.76	Cinchonain (VII of VII)	C_24_H_19_O_9_	451.1039	−1.915	341, 289, 271

**Table 2 plants-11-01737-t002:** Physical characteristics of nanoformulations F-1 and F-2.

Sample	Size Average (nm)	Polydispersity Index (PDI)	%EE
F-1	26.1 ± 0.8	0.463 ± 0.025	92 ± 3
F-2	112.0 ± 3.0	0.368 ± 0.021	98 ± 1

**Table 3 plants-11-01737-t003:** Antioxidant activity of free PA-E and nanoencapsulated formulations obtained from *U. tomentosa* leaves.

	IC_50_ (µg/mL) ^1,2^			
	Ethanol ^3^	Water ^3^	F-1 ^3,4^	F-2 ^3,4^
PA-E	13.6 ^a^ ± 0.1	41.2 ^b^ ± 2.0	13.5 ^a^ ± 0.1	12.5^a^ ± 0.8

^1^ IC50 µg/mL of sample. ^2^ Values are expressed as mean ± standard deviation (S.D.). ^3^ Different superscript letters in the same column indicate differences are significant at *p* < 0.05 using one-way analysis of variance (ANOVA) with a Tukey post hoc. ^4^ Samples of nanoparticles for the evaluation of antioxidant activity were prepared in water.

**Table 4 plants-11-01737-t004:** Delayed-type hypersensitivity (DTH) test expressed as change in footpad thickness for each treatment measured at 0, 24, 48 and 72 h.^1^.

Formulation	0 h (mm)	24 h (mm)	48 h (mm)	72 h (mm)
Water	0.01 ± 0.03 ^a^	0.19 ± 0.05 ^a^	0.11 ± 0.07 ^a,b^	0.07 ± 0.02 ^a,b^
PA-E	0.00 ± 0.01 ^a^	0.19 ± 0.09 ^a^	0.15 ± 0.07 ^b,c^	0.11 ± 0.06 ^a^
F-1	0.00 ± 0.01 ^a^	0.20 ± 0.06 ^a^	0.23 ± 0.07 ^c,d^	0.12 ± 0.06 ^a^
F-2	0.00 ± 0.01 ^a^	0.26 ± 0.08 ^a^	0.25 ± 0.04 ^d^	0.08 ± 0.05 ^a,b^

^1^ Values are expressed as mean ± Standard Deviation. Different superscript letters in the same column indicate differences are significant at *p* < 0.05 using one-way analysis of variance (ANOVA) with a Tukey post hoc.

## Data Availability

Not applicable.

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
