# Peer review of "Hybrid Nanoparticles of Proanthocyanidins from *Uncaria tomentosa* Leaves: QTOF-ESI MS Characterization, Antioxidant Activity and Immune Cellular Response"

_plants, 2022, doi:10.3390/plants11131737_

Round 1

Reviewer 1 Report

Articles are well written. Can be accepted after minor revision

Abstract need some modification. Rewrite the abstract in scientific manners.

Discussion part should me more elaborated and should be according to results. 

Author Response

Abstract has been rewritten in a scientific manner and discussion has been improved.

Reviewer 2 Report

The manuscript “Proanthocyanidins QTOF-ESI MS characterization, Antioxidant activity and Delayed-type hypersensitivity reaction augmented by Hybrid Nanoparticles of Uncaria tomentosa leaves from Costa Rica” by Araya-Sibaja et al. describes the characterization of Proanthocyanidins of Uncaria tomentosa leaves and their antioxidant performance once loaded into lipid and proteic nanoparticles. The work is very well designed and described. Results are clearly reported.  I only suggest to

  • correct all errors in table 2 and 3: i.e. 11.80 ± 3.27 should be written as 12±3 or 26.10 ± 0.82should be written as 26.1±0.8, etc
  • Correct also the value in Table 2 : 111.80 should be 11.80
  • Line 52: add the term activities
  • line 296: are instead of and
  • Please reformulate sentences: line 275-276 and 303-304
  • In IR discussion use the term band instead of peak

Author Response

All the suggestions have been attended to accordingly.

Reviewer 3 Report

The article is well written and is suitable for publication in its current form

Author Response

Thank you for the comments.